# Effect of Leaching Behavior on the Geometric and Hydraulic Characteristics of Concrete Fracture

**DOI:** 10.3390/ma15134584

**Published:** 2022-06-29

**Authors:** Yuan Wang, Mengmeng Tao, Di Feng, Yu Jiao, Yulong Niu, Zhikui Wang

**Affiliations:** 1Department of Water Conservancy and Hydropower Engineering, Hohai University, Nanjing 210098, China; wangyuan@hhu.edu.cn; 2Department of Civil and Transportation Engineering, Hohai University, Nanjing 210098, China; fengdi@126.com (D.F.); jiaoyu0125@163.com (Y.J.); wangzhikui23@163.com (Z.W.); 3China Three Gorges Corporation, Beijing 100038, China; niuyl89@163.com

**Keywords:** underground engineering, leaching of concrete, fracture geometric characteristics, hydraulic characteristics

## Abstract

The leaching of material from concrete fracture surfaces has an impact on the structural concrete in service, but the number of studies that consider the effect of the coupling of the leaching, fracture geometry and hydraulic processes on concrete fractures is insufficient. In this study, a series of experiments was conducted, and a leaching model proposed, to investigate the mechanism of leaching behavior on the geometric and hydraulic characteristics of concrete fractures. Following the leaching experiment, the evolution of fracture geometric characteristics was observed by a three-dimensional (3D) laser scanning technique, finding that the fracture produces residual leached depth and local uneven leaching, which results in a decrease in roughness. The hydraulic characteristics were then investigated by permeability tests, and it was found that the fracture hydraulic aperture and permeability increase monotonically with leaching time. A simulation of fluid flow in a numerical fracture revealed the effect of residual leached depth and a decrease in roughness on the hydraulic characteristics. Finally, based on the analysis of the chemical composition of the leaching solution, a leaching model of concrete rough fracture surface is proposed and the mechanism of leaching behavior is discussed. These new findings are useful for the understanding of the development of leaching, local to concrete fracture surfaces.

## 1. Introduction

Concrete is an excellent building material with high strength and reliable water resistance; it is widely used in underground engineering [1], such as in tunnel lining [2] and coal mine shafts [3]. The phenomenon of leaching consists in the dissolution of solid calcium in cement hydrates when concrete is exposed to any aggressive solution (most of the time, pure water or water with a very low calcium concentration) [4,5,6]. This leaching process involves the dissolution of the most soluble phase of cement hydrates, calcium hydroxide (Ca(OH)_2_), and the subsequent transport of dissolved ions out to the environment [7,8,9]. The long-term effect of this leaching phenomenon is to weaken the material’s solid matrix and the concrete’s durability [6,7,10,11], thereby causing the degradation of concrete structures in aggressive environments [12]. At the same time, structural concretes in service develop fracturing from different causes, including early-age thermal shrinkage or long-term mechanical loadings [2]. Fractures provide preferential transport pathways for the ingress of water, which contributes greatly to the effect of leaching of hydration products from fractured regions in concrete [13,14]. Therefore, studying the leaching of hydration products from fractured regions in concrete has more important engineering significance than concrete itself [7,15] (Figure 1).

Compared with the relatively flat surface of concrete, the surface of fractures in concrete is rough, and the geometric characteristics are complicated [16,17]. The roughness and complicated geometric characteristics affect the hydraulic characteristic of fractures (hydraulic aperture) [18,19,20,21]; the hydraulic characteristic controls the flow state of water in fractures [22] and the flow state, in turn, affects the leaching characteristics [7]. Leaching also alters the fracture geometric characteristics [23,24], and the evolution of fracture geometric characteristics then affect the characteristics of fracture space [25]; this determines the hydraulic characteristics of fractures [26], and the evolution of hydraulic characteristics will further affect the leaching characteristics on the surface of concrete fractures [13]. This is a coupling of the leaching, fracture geometry and hydraulic processes, shown in Figure 2. The effects of fracture geometric characteristics on hydraulic characteristics are many. Brown et al. [25] considered the roughness of fracture surface and modified the cubic law to obtain the hydraulic aperture. Zoorabadi et al. [27] established a new equation between mechanical and hydraulic aperture for different roughnesses. Chen et al. [28] studied the effect of fracture geometric characteristics on the permeability in deformable rough-walled fractures. In contrast, the effect of leaching on the fracture surface geometric characteristics was only explored by a few [29]. Recently, however, it has gradually attracted the attention of several scholars. For example, Wang et al. [30] and Duan et al. [31] studied the effect of leaching on a series of geometric characteristics of limestone fractures. Compared with limestone, concrete is an artificial material composed of fine and coarse aggregates and additives [6], and its physical and chemical properties are quite different. The evolution of fracture geometric and hydraulic characteristics after concrete fracture leaching was not well investigated in previous studies.

In summary, it is necessary to study the evolution law of fracture geometric characteristics after the leaching of concrete, and its effect on hydraulic characteristics. In this paper, the direct leaching method was used to carry out leaching experiments on concrete fractures; several three-dimensional (3D) laser scanning tests and permeability tests in different time periods were then conducted to observe the evolution of fracture geometric characteristics and their effects on the hydraulic characteristics. Numerical simulations were compared to reveal the effect of variation in fracture surface geometric characteristics on hydraulic characteristics. Finally, the chemical composition of the leaching solution was analyzed and a leaching model of rough fracture surface is proposed to discuss the mechanism of leaching on concrete rough fracture surface. The results of this research paper will contribute, to some extent, to a better understanding of the development and mechanism of leaching on concrete fractures.

## 2. Materials and Methods

In an attempt to approach a real rough fracture, the Brazilian splitting test was used to make a random fracture surface. A series of leaching experiments were carried out on concrete samples with single fractures. In the experiments, the fracture surface geometric characteristics were obtained by 3D laser scanning technology and the fracture permeability was tested using a permeability experimental setup every 120 h to observe the evolution law of leaching on the fracture surface geometric characteristics and the influence of this evolution on fracture permeability, respectively. To validate universal conclusions, two groups of reproducibility experiments were added.

### 2.1. Preparation of Fracture Samples

First, concrete samples with rough single fractures were prepared before the experiments. We referred to the method of preparing concrete samples with rough single fractures introduced by Anwar [14]. Firstly, the concrete was configured and poured into concrete cylinders. The Brazil splitting test was then used to produce rough single fractures. The main steps were as follows:(1)Configuring the concrete. In order to meet the strength and water resistance requirements of general underground engineering [32], composite cement P.C.32.5 and medium sand (fineness modulus between 2.3 and 3.0) were selected and mixed with clean water. The mixing ratio of cement, sand and water was 1:3:0.55. A waterproof agent (SJM-1500, Suzhou Institute of Building Science Group Co.,Ltd., Suzhou, China) was added to reduce the permeability of the concrete (2% of cement dosage).(2)Pouring the concrete cylinders. A release agent was applied on the inner wall of each cylindrical mold (height 50 mm and outer diameter 50 mm) to facilitate demolding after initial setting. The prepared concrete was poured into the molds three separate times. After each pouring, each mold was placed on the shaking table and shaken for 30 s. The purpose of shaking was to eliminate the bubbles in the concrete to make the filler denser. The shaking time was controlled to prevent the separation of solids and liquid in the concrete. After all the concrete was poured, the concrete-filled molds were put into the curing room at a relative humidity of 100% and temperature of 20 °C for the initial setting of the concrete. When the initial setting of the concrete was completed (about 8 h), each mold was disassembled and the concrete cylinder removed.(3)Curing the concrete. The intact cement cylinders were placed in the curing room for further curing and left for 28 days to complete the curing process.(4)Making the rough single fracture. A specially designed Brazilian splitting test machine was used to split each cylinder into two half-cylinders along the long axis, creating an artificial fracture by tensile stress within the sample.

Six rough fracture surfaces on three pairs of samples were processed in the same way. Under the same experimental conditions, the three pairs of samples were considered as reproducibility experiments to obtain the universal law. The three pairs of samples were named S1, S2 and S3, respectively, and the suffixes A- and B-side were added to the two halves of each pair of samples, respectively (see Figure 3). For example, S1A indicated the A side of sample S1.

### 2.2. Leaching Experiments

The direct leaching method enables the placing of the cement-based material samples (cement, mortar, concrete, etc.) in an aggressive solution (deionized water, mineral water, sulfate, etc.) [33,34]. If the actions of water head and mechanical stress are not to be considered, the direct leaching method is the most widely used contact leaching method because of its simplicity [33]. In this paper, the effect of water head and mechanical stress were not considered, so the direct leaching method was selected. The six rough fracture surfaces from three pairs of samples were put in a constant temperature water tank (25 °C). Deionized water was selected as the aggressive solution because deionized water, itself, has good leaching ability, and was often used as the aggressive solution in leaching experiment by other scholars [5,6,10]. Based on the experimental experience of Duan et al. [31], the deionized water was replaced every 24 h in order to simulate the flow of the groundwater environment and maintain a high concentration gradient.

### 2.3. Evolution of Fracture Surface

#### 2.3.1. Obtainment of Fracture Surface

3D laser scanning technology is a non-contact three-dimensional measurement technology which can quickly obtain a wide range of high-precision fracture surface geometries without damaging the fracture surface [28]. The scanning interval used in this paper was 0.05 mm and the scanning accuracy was 0.01 mm, which satisfied the requirements of obtaining the geometric characteristics of a fracture surface [35,36]. In order to control the same fracture surface scanned at different times in a unified coordinate system, identifiable coordinate marks were marked on the outer surface of each sample (see Figure 4).

As shown in Figure 5, red indicates high elevation and blue indicates low elevation. It can be seen that the fracture surface has a complicated geometry, from which key geometric characteristics must be extracted and characterized for further study.

#### 2.3.2. Analysis of Geometric Characteristics of Fracture Surface

After obtaining the fracture surface geometry in Section 2.3.1, the geometric characteristics were analyzed. Based on previous studies [20,25] on the relationship between fracture geometric characteristics and hydraulic characteristics, roughness is an important indicator affecting hydraulic characteristics [20]; it was, therefore, first necessary to accurately identify the surface roughness [37]. The joint roughness coefficient (JRC) is a basic parameter proposed by Barton [38] to describe the roughness fluctuation of a fracture surface and is widely used in engineering [39]. The JRC values can be calculated with a dimensionless parameter *Z*_2_ [40], widely discussed, and defined as follows [40,41]:(1)$$Z_{2}=\frac{1}{L}\overset{x=L}{\underset{x=0}{\int }}{\left(\frac{dz}{dx}\right)}^{2}=\sqrt{\frac{1}{n}\overset{n}{\underset{i=1}{\sum }}{\left(\frac{Z_{i+1}-Z_{i}}{X_{i+1}-X_{i}}\right)}^{2}}$$

Subsequently, the JRC values can be evaluated by [40]:(2)$$JRC=32.2+32.47lgZ_{2}$$
where, *L* is the nominal length of the profile, *X_i_* is the ith segment of *L*, *Z_i_* is the amplitude of the roughness of the profile. According to the definition of the JRC, it is initially the quantization parameter of the profile (2D). The JRC values of all profiles on the same fracture surface by scanning interval can be calculated and averaged to characterize the JRC value of the fracture surface (3D).

It can be seen from Table 1 that the JRC values of the fracture surfaces of the A- and B-side from the same pair of samples are almost the same, but not exactly consistent. This is because the fracturing process of concrete is often accompanied by particle disintegration, so the two surfaces of the A- and B-side cannot be completely consistent, and the JRC values are not exactly the same.

According to Equations (1) and (2) and the calculation method of the 3D fracture surface JRC, the JRC value of the 3D fracture surface represents the macro roughness of the fracture surface. In order to obtain a detailed description of the fracture surface at the meso level, additional parameters describing the geometric characteristics were needed. Therefore, the distribution of slope [42] was added to the description of the fracture surfaces at the meso level. The slope was defined as follows [42]:(3)$$Slope=\frac{dz}{dx}=\frac{Z_{i+1}-Z_{i}}{X_{i+1}-X_{i}}$$

It can be seen that the slope is one segment unit in the JRC calculation, which represents the rough fluctuation of a segment unit in the fracture surface. Correspondingly, the JRC is the averaging of the slopes of all segment units on the fracture surface. Therefore, the JRC and slope distribution can describe the roughness characteristics of the fracture surface more comprehensively at the macro and meso levels, respectively. Figure 6 shows the slope distribution of the fracture surfaces, where black represents the A-sides and red represents the B-sides.

Figure 6 shows that the slope distribution of the fracture surfaces exhibit Gaussian distribution. The slope of S1 is −0.037 ± 0.485 (for S1A) and 0.019 ± 0.436 (for S1B). The slope of S2 is −0.002 ± 0.433 (for S2A) and −0.002 ± 0.422 (for S2B). The slope of S3 is 0.020 ± 0.434 (for S3A) and −0.028 ± 0.450 (for S3B).

### 2.4. Permeability Test of Single Fracture

For the purpose of achieving the hydraulic characteristics of the fractures, we used a self-developed permeability experimental setup to test permeability of fractures (see Figure 7). The cell pressure (i.e., confining pressure) range was 0–80 MPa and the accuracy was 0.01 MPa. The water pressure difference (i.e., pressure difference between inlet- and outlet- pressure) range was 0~16 MPa and the accuracy was 0.01 MPa. These pressures were controlled by computer. The concrete sample was saturated according to the operational standard in order to remove air from the samples. After gently washing the fracture surfaces with deionized water, the two half-concrete samples were fitted carefully together to avoid introducing small debris into the fracture. Samples were marked to ensure that the two half-concrete samples matched equally between different experiments. The fitted sample was then confined within the rubber membrane by iron hoops. Filter paper was sandwiched between the sample and the cushion block to prevent small debris from blocking the outlet pipe, and the sample was then placed into the pressure cell (see Figure 8). Each concrete sample with a single fracture was subjected to a confining pressure of 2 MPa and a water pressure difference of 0.2 MPa, and monitored by computer to accurately maintain the specified pressure. The measurable flow rate was obtained in the process. The only liquid used in the permeability experiments was also deionized water.

The linear Darcy’s law was used to describe the laminar fracture flow at low velocity, given by [43]:(4)$$k=\frac{\mu QL}{A\nabla P}$$
where *k* is the intrinsic permeability, *μ* is the fluid viscosity, *Q* is the flow rate, *A* is the cross-sectional area, and ∇*P* is the differential pressure. Similarly, the fracture permeability was evaluated via the hydraulic aperture obtained using a parallel plate approximation, namely [44]:(5)$$b_{n}={\left(\frac{12\mu LQ}{W\nabla P}\right)}^{1/3}$$
(6)$$k=\frac{b_{n}^{2}}{12}$$
where *b_n_* is the hydraulic aperture and *W* is the sample width.

## 3. Results and Analyses

### 3.1. Variations in the Fracture Geometric Characteristics

Through the techniques and methods introduced in Section 2.3, the evolution results of the geometric characteristics of concrete fracture surfaces after the leaching experiments were summarized in universal laws and analyzed for causes.

#### 3.1.1. Evolution of Fracture Surface Geometry

First, since the evolution laws are consistent, the S2 fracture surface was selected as an example to show its variation in geometry.

In each group surface variation shown in Figure 9, the upper half is the initial fracture surface before the leaching experiment (0 h), and the lower half is the fracture surface after the experiment (480 h). The middle part shows the intact fracture surface, and the left and right parts are enlarged images of dotted boxes to facilitate the observation of detailed geometry. The legend indicates the elevation of the fracture surface. The red wireframe in the figure shows the local areas before the experiment, and the white (or black) wireframe shows the changed area after the experiment. It can be seen that the red areas (several areas with high elevation) decrease in size and fade in color after leaching, while the blue areas (several areas with low elevation) increase in size and deepen in color after leaching. This shows the degraded depth of the fracture surface elevation, the surface solid components of the fracture surface (such as calcium hydroxide) being removed by the leaching experiment; that is, the leaching phenomenon was confirmed in the fracture surface geometry.

Second, in order to further obtain the degraded depth of the fracture surface elevation and summarize the universal law, the elevation distributions of all fracture surfaces are shown in Figure 10, where, the black line indicates the distribution of elevation before the leaching experiment and the red line indicates the distribution after the leaching experiment.

It can be seen from Figure 10 that the peak value of the elevation distribution curve of all fracture samples moves to the left; that is, the macroscopic elevation of the fracture surface shows a declining law, which proves the results shown in the 2D cloud above (see Figure 9), and also shows that the influence of leaching on the morphology of the fracture surface has a universal law. The degraded depth of the fracture surface is called the leached depth [5,10].

#### 3.1.2. Evolutions of Fracture Surface Macro Roughness

In order to study the evolution of fracture surface roughness, the JRC values of three pairs of samples at 0 h, 120 h, 240 h, 360 h and 480 h are listed in Table 2.

It can be seen from Table 2 that the JRC value of the fracture surface in each test decreases with the increase in leaching time. It is not appropriate to compare the JRC between different tests in Table 2, because the concrete samples are different in each test. Therefore, in order to compare the sensitivity of leaching time to roughness, the ratios of the JRC relative to initial conditions (0 h) (i.e., normalized JRC) are calculated and shown in Figure 11.

Figure 11 shows that S1A decreased by 3.97%, S1B decreased by 4.74%, S2A decreased by 6.83%, S2B decreased by 5.45%, S3A decreased by 4.35% and S3B decreased by 2.71% from 0 h to 480 h. These findings are consistent with those of Duan et al. [31] in that the roughness of the limestone fracture shows a decreasing trend after long-term leaching in deionized water.

It can be seen from Figure 10 that although there are exceptions (S2B and S3B), the elevation of fracture surfaces decreases unevenly and shows a more centralized trend. The peak value of fracture surface elevation distribution after the leaching experiment moves to a lower elevation than that before the experiment. The peak value is larger, that is, the frequency is larger. The centralization of elevation indicates that the fracture surface elevations decrease unevenly, but some of the higher elevations decrease more and some of the lower elevations decrease less. Therefore, the fracture surface tends to be flattened, with mainly the raised part of the fracture being flattened. According to the meaning of roughness (JRC), when the fracture surface geometric characteristics tend to be flat, its roughness decreases. This explains very well the continuous decrease in JRC seen in Figure 11. The evolution in roughness (JRC) is still at the macro level, and we need to continue to investigate the mechanism of the effect of leaching on the fracture surface characteristics at the meso level of the detailed part of the fracture surface.

#### 3.1.3. Evolutions of Fracture Surface Local Details

In order to further study the evolution of the fracture surface details at the meso level and investigate the mechanism of the uneven variation of the fracture surface elevation, we first studied the evolution of the slope distribution of the fracture surface, as shown in Figure 12.

Where, black indicates the slope distribution of the fracture surface before the experiment, red indicates the slope distribution after the experiment, and the arrows point to the evolution trend in the slope distribution. It can be seen in Figure 12 that the area with a larger absolute value of slope on the fracture surface of all samples decreases and the area with a smaller absolute value of slope increases; furthermore, the slope distribution after the leaching experiment is more centralized towards the slope with a lower absolute value. Further proof can be obtained from the standard deviation of the slope distribution of the fracture surface from the start to the end of experiment. The standard deviation of S1A decreases from 0.485 to 0.455, that of S1B decreases from 0.436 to 0.407, that of S2A decreases from 0.433 to 0.392, that of S2B decreases from 0.422 to 0.391, that of S3A decreases from 0.434 to 0.407, and that of S3B decreases from 0.450 to 0.432. The lower the standard deviation, the more centralized the distribution. From the centralization of slope distribution, it can be inferred that asperities with higher slopes were flattened to lower slopes by the leaching effect.

Finally, to verify this mechanism more visually, we fit the two fracture surface models from same pair of samples. According to the triangular stability principle, ignoring the deformation of fracture surface, it is assumed that there are only three contact points between the two fracture surfaces which intercept the profile along the seepage direction (see Figure 13).

The profile of S2 at Y = 10 mm was selected as an example for presentation and analysis.

It can be seen from Figure 14 that after the fracture surfaces of the same sample were fitted together, the initial profiles of A- and B-side surfaces are not exactly coincident (as outlined by the solid line in the figure), which is also consistent with the phenomenon that the JRC values of the A- and B-side surfaces are not same (see Table 1); this proves the existence of an initial aperture in the fracture, providing a channel for fluid flow. The profiles are shifted backward after being subjected to the leaching experiment (with the A-side surface at the bottom, the profile is shifted downward; with the B-side surface at the top, the profile is shifted upward), which is consistent with the evolution law of the elevation decrease of the fracture surface (leached depth) in Figure 10. However, the average value of the backward shift of the fracture profile is 63.8% of the average of leached depth. The leached depth is produced by the leaching of the cement component in the concrete, which also leads to the phenomenon of the profile on the same side shifting backward (solid to dashed line). It should be noted that when the fracture surface is fitted, especially when the permeability is being measured (the confining pressure is applied), the two surfaces of the A- and B-side will be in contact with each other. Therefore, the degraded depth of the fracture surface evolution (leached depth) is not completely consistent with the backward distance of the fracture profile after fitting. However, the backward distance will not be completely cancelled because concrete is a special material with a combination of many components. As mentioned in Section 2.1, regarding the preparation of samples, concrete mainly includes cement, water and sand particles, so sand particles and gravel will be embedded into the fracture surface of concrete (as shown in Figure 15). The leaching property of a sand particle is different from that of cement. The property of a sand particle is more stable, the leaching rate is low, and it is not easily dissolved. Therefore, when fitting fracture surfaces, these sand particles will become the support of the aperture and prevent the fracture surface from getting bigger. In terms of the profile, although there is backward distance, the amount of backward distance is not as large as the leached depth. We, therefore, refer to this backward distance as the residual leached depth after the sample is fitted.

It can be seen from the enlarged view (see Figure 14b,c) that, in addition to the residual leached depth of the profile, the profile local asperity with a large slope becomes smoother and the slope decreases after the leaching experiment. This directly proves the inference that the large slope of the asperity is flattened into a low slope, and this is caused by the leaching experiment; it also reveals the direct reason for the decline law of the JRC value of fracture surface. Moreover, it intuitively shows the evolution of the slope distribution in Figure 12, and further verifies the analysis of the evolution law of fracture surface elevation.

So far, the evolution law of fracture surface geometric characteristics has been observed and analyzed at the macro level (JRC) and the meso level (slope distribution and profile).

### 3.2. Variations in Fracture Permeability

In the permeability experiment, the samples were tested for different leaching times (0 h, 120 h, 240 h, 360 h and 480 h). In order to obtain the universal law, three groups of tests with the same conditions were carried out, specifically, with a confining pressure of 2 MPa and a seepage pressure difference of 0.2 MPa. The direct results of the experiment, the steady flow rate, are recorded in Table 3.

It is seen in Table 3 that, under the same conditions (confining pressure 2 MPa, seepage pressure difference 0.2 MPa) after the leaching experiment, the flow rate in the fracture increases. S1 increases by 146.4%, S2 increases by 98.6% and S3 increases by 140.7%. The hydraulic aperture (*b_n_*) and permeability (*k*) of each group of samples, according to Equations (1)–(3) introduced in Section 2.4, are shown in Table 4 and Table 5, respectively.

Similar to the normalized JRC in Section 3.1.2, it is not appropriate to compare the hydraulic characteristics between different tests in Table 4 and Table 5, because the concrete samples have different initial mechanical apertures. Therefore, in order to compare the sensitivity of leaching time to the hydraulic characteristics, the ratios of hydraulic aperture and permeability, relative to initial conditions (0 h), i.e., normalized hydraulic aperture and normalized permeability, were calculated and are shown in Figure 16.

It can be seen from Figure 16 that the curves for the evolutions of hydraulic aperture and permeability present the same shape, but the increase rate of permeability is much higher than that of hydraulic aperture. As the leaching time increases from 0 to 480 h, the hydraulic apertures of S1, S2 and S3 increase by 35%, 26%, 34%, while the permeability of S1, S2 and S3 increases by 82%, 58%, 80%, respectively. Since fracture is the main channel of fluid flow, combined with the evolution of fracture characteristics in the Section 3.1, it can be considered that the evolution of fracture hydraulic characteristics is closely related to the evolution of fracture geometric characteristics. The evolution of the geometric characteristics of the fracture surface after the leaching experiment was summarized and analyzed in Section 3.1, and the concept of residual leached depth was presented. The generation and development of the residual leached depth widens the seepage channel of the fracture and increases the hydraulic aperture. Meanwhile, the roughness and slope of the fracture surface also changes (the JRC value decreases and the slope distribution concentrates to a low absolute value) due to the local asperity of the fracture surface being flattened, caused by the leaching experiment. This will weaken the resistance effect of a rough sidewall on the fluid flow in the fracture, further improving the permeability of the fracture, and increasing the hydraulic characteristics of the fracture.

In order to validate this analysis, the fluid in the fracture was simulated on the profile (S2, Y = 10 mm), shown in Figure 14.

Incompressible Newtonian flow is governed by the well-known Navier–Stokes equations [43,45,46]:(7)$$\rho \left(\frac{\partial U}{\partial t}+U\cdot \nabla U\right)=-\nabla P+\mu \nabla^{2}U+F$$
where *ρ* is the fluid density, *U* is the velocity vector of flow particle and *F* is the body force vector. Because the Navier–Stokes equation cannot be solved directly, the numerical method is widely used to solve it [47,48,49], and to simulate and study the flow characteristics of a fluid in rough fractures [50]. In this paper, Fluent 16.0 was used to simulate and analyze the seepage in rough fractures. Many scholars [46,51,52] verified the reliability of the calculation software. The inlet on the left-hand side of the fracture model was set as the velocity inflow boundary: the velocity magnitude was 1 m/s and the free outlet boundary was on the right-hand side. The upper and lower boundaries of the fracture model were set as the wall boundary conditions without fluid flow or slip.

Figure 17a,b shows the velocity distribution in the X direction of the fluid in the fracture before and after the leaching experiment. The seepage channel has been widened, and the hydraulic aperture and permeability have increased. The simulation results show that the hydraulic aperture increased by 82% and the permeability increased by 231%. Figure 17c,d is a local enlarged view, showing the seepage at the asperity where the local slope decreases at X = 17−20 mm. It can be seen that the resistance of the sidewall to the fluid is weakened.

Since Figure 17 shows the actual profile of fracture before and after leaching experiment, both the mechanical aperture and roughness are variables. Therefore, in order to further analyze the influence of the evolution of roughness on the fluid in the fracture, the profile at Y = 10 mm and X = 17–20 mm on S2A, before and after leaching, were selected as the wall boundaries with different roughnesses, respectively; the profiles were horizontally translated by the same 0.2 mm aperture to establish an idealized fracture model. In this model, the influence of varying roughness on fracture seepage can be studied.

It can be seen from Figure 18 that the streamline in Figure 18a is significantly more tortuous than that in Figure 18b, and the streamline at the center of the aperture in Figure 18a is 3.65% longer than that in Figure 18b. The simulation results show that the hydraulic aperture of Figure 18b increases by 46% and the permeability increases by 112% compared with Figure 18a. This shows that when the mechanical aperture is the same, the decrease in fracture surface roughness leads to the increase in fracture permeability. The numerical simulation results are consistent with the experimental results, and validate the analysis of the evolution mechanism of fracture hydraulic characteristics well. Moreover, the results of the experiment and the numerical simulation agree well with the empirical equation according to a large number of experiments presented by Barton et al. [20,53]:(8)$$b_{n}=\frac{b_{m}^{2}}{JRC^{2.5}}$$
where *b_m_* is mechanical aperture. Equation (8) shows that in the case of a constant mechanical aperture, as the JRC increases, the hydraulic aperture also increases.

## 4. Discussion

### 4.1. Mechanism of Rough Fracture Surface Leaching

To further investigate the mechanism of the effect of leaching on the geometric characteristics of the fracture surface, we measured the elements and concentrations of the deionized water, soaked during the leaching experiment, by inductively coupled plasma mass spectrometry (ICP-MS). Since the initial deionized water contained no other element, the measured element concentrations should originate from the concrete samples only. Figure 19 shows the chemical composition analysis results and pH values for a period of time after the replacement of the deionized water.

From the graph, it can be seen that the calcium ion concentration changes to the greatest extent and is consistent with the pH trend (R = 0.957). Referring to previous studies [4,11,54], it can be assumed that the main chemical reaction occurring in concrete leaching experiment is:(9)$$Ca{\left(OH\right)}_{2}=Ca^{2+}+2{\left(OH\right)}^{-}\;$$

That is, mainly the cement components in concrete undergo a leaching reaction, so the following calcium concentration is the main object of study. Figure 19 shows that the gradient of the calcium concentration curve is initially large, then small and, finally, stabilizes when it reaches 60–70 min. It indicates that the leaching reaction rate is from fast to slow and reaches the near equilibrium state at the later stage. The mechanism of the leaching reaction can be inferred from this. The initial deionized water has a low calcium concentration, so the gradient in calcium concentration between the water and the fracture solid surface is large, and the rate of leaching is fast. As the leaching reaction proceeds, the calcium concentration continues to rise, and the calcium concentration reaches a higher level at the later stage (after 60 min) and remains stable: the leaching reaction is basically balanced, and the PH value also tends to be stable. When the calcium concentration in deionized water is very low, the calcium concentration gradient is large, which is conducive to driving the rapid precipitation reaction of calcium hydroxide components on the concrete surface. When the calcium concentration in the leaching solution increases and the gradient of calcium concentration decreases, the leaching reaction rate of calcium hydroxide decreases and the reaction between solid and liquid reaches equilibrium, inhibiting leaching from proceeding. Thus, the existence of the calcium concentration gradient between the concrete fracture solid surface and the leaching solution can be considered as the mechanism of leaching reaction, which agrees well with the mainstream view [13,33].

Based on this theory, the mechanism by which the geometric characteristics of rough fracture surfaces is affected by leaching is further discussed. A rough fracture surface leaching model is proposed based on the profile (see Figure 14), as shown in Figure 20.

In Figure 20, the dashed line is the initial profile; the solid line is the final profile after the leaching experiment; the blue circles represent ions or ion clusters (mainly calcium); the size and density indicate ion mass and concentration; the dashed line with arrows indicates the direction of ion transport; and the length of the dashed line indicates the rate of ion transport. Figure 20 shows the asperity in the slope of the raised part of the wide surrounding area; the nearby deionized water flow is great so the leached calcium ions are easily diffused and carried away by the water flow. Therefore, the concentration of calcium ions in the deionized water around the raised part will maintain a low value, and as the ion mass and concentration around the raised part in the model is small, this region maintains a higher gradient in calcium concentration. This, in turn, promotes the raised asperity part to leach further, and the ion transport rate becomes greater, as can be seen in the model. While the asperity in the slope of the depressed part is limited by the surrounding raised part, mobility is poor. Moreover, early by the higher initial calcium concentration gradient effect results in the leaching of calcium ions. Calcium ions tend to aggregate, so the concentration of calcium ions in deionized water around the depressed part will maintain a higher value. As the model around the depressed part of the ion mass and concentration is larger, and this region maintains a lower calcium concentration gradient, it inhibits the depressed asperity part from further leaching, so the ion transport rate in the model is smaller. In summary, the leaching in the raised part is promoted and the leaching in the depressed part is inhibited, so the leaching in the raised part is faster and greater than the leaching in the depressed part. After the leaching experiment, the profile of the raised part moved back more than that of the depressed part (see Figure 14b,c), resulting in the decline of the local slope, that is, the evolution law shown in the slope (see Figure 11). When the whole rough fracture surface is affected by this leaching mechanism, it leads to the reduction of roughness, that is, the evolution law shown in the roughness (see Figure 10). This model reveals effectively the mechanism of the geometric characteristics of the rough fracture surface under the action of leaching at the micro level.

### 4.2. Mechanism of the Effect of Leaching on Fracture Hydraulic Characteristics

In this paper, we found that fracture permeability increases gradually with the development of leaching. In the previous sections, we investigated the impact of leaching on the geometric characteristics of fracture surfaces, which leads to the evolution of fracture hydraulic characteristics. That is, the rough fracture surface is leached, and the surface elevation decreases; the residual leached depth is generated after the samples were fitted together, resulting in an increase in aperture. Moreover, the roughness of the fracture surface (JRC) decreases and the effect of inhibiting flow also decreases. These actions together lead to the increase in the hydraulic aperture and the permeability of fractures.

However, the law that the decrease in fracture surface roughness leads to the increase in permeability cannot be simply understood as the lower the roughness, the higher the permeability, especially between different samples. We observed that the initial average JRC of S1, S2 and S3 on both surfaces of the A- and B-sides are 21.3, 20.2, 20.7, but the hydraulic aperture is 1.55 × 10^−5^ m, 1.49 × 10^−5^ m, 1.44 × 10^−5^ m, respectively, and the permeability is 2.01 × 10^−11^ m^2^, 1.84 × 10^−11^ m^2^,1.73 × 10^−11^ m^2^, respectively. The permeability of S2, which has the smallest JRC, is instead smaller than that of S1, which has the largest JRC. This is because the initial mechanical aperture between them is different (see Equation (8)). Furthermore, the JRC values of the three groups of samples are very close to each other, which makes the influence caused by other chance factors very large, so it is difficult to establish an accurate relationship between the JRC value of a fracture surface and its hydraulic characteristics (hydraulic aperture and permeability). In this study, the main purpose of designing three pairs of samples was to conduct a repeated experiment from which to derive a universal law governing the effect of leaching on the fracture surface. The relationship between the roughness JRC and the hydraulic characteristics obtained is, therefore, the only law presented by the same sample under the effect of leaching. The inability to establish an exact equation for the relationship between the JRC and hydraulic characteristics does not prevent the correctness of the conclusion that the decrease in fracture surface roughness caused by leaching leads to an increase in permeability. This is because it is based on the law observed for the same pair of fracture surfaces at different leaching times and is verified by repeatability experiments and analyzed by numerical simulations.

## 5. Conclusions

In this study, leaching behavior acting on the fracture was verified through laboratory work. The main conclusions are summarized as follows:(1)The series of 3D laser scanning test results show that the elevation of fracture surfaces decreases unevenly, the JRC decreases monotonically with leaching time and the slope distribution centralizes to a low absolute value. The solid element of the concrete fracture surface was leached by an aggressive solution (deionized water) and the degree of this leaching is uneven in different areas of the fracture surface. The leached degree of the high slope area is greater than that of the low slope area, and this analysis was verified from the profile.(2)The hydraulic characteristics and the fluid flow state in the fractures were investigated by using the permeability test combined with the fracture numerical model. The existence of residual leached depth widens the hydraulic aperture of the fracture, and the flattened asperity weakens the resistance to the fluid. This variation in geometric characteristics, caused by leaching, increases the hydraulic aperture and permeability of fractures.(3)The chemical analysis of the soaked solution in the leaching experiment showed an inevitable linkage between the leached degree and the calcium concentration gradient in the solution. Therefore, a leaching model of concrete rough fracture surface was proposed to describe the mechanism of leaching on the fracture characteristics. The rough surface of the fracture leads to an uneven calcium concentration gradient near the asperity, resulting in a greater degree of leaching in the raised part than that in the depressed part.

This study confirms that the leaching behavior acting on a natural rough concrete fracture exposed to aggressive solution results in a long-term evolution of geometric and hydraulic characteristics. This conclusion is useful for a better detection of the variation in fracture of structural concrete in service from the standpoint of practical applications and for further illustration of hydraulic performance in the long term.

The leaching method used was relatively simple and the proposed model is straightforward because the interactions of normal pressure in the leaching processes are omitted; these are, however, applicable to open fracture without normal pressure in concrete structures. Generally, when considering a closed fracture under normal pressure, the phenomenon of pressure leaching may occur and be significant [44,55,56,57]. This phenomenon will make the concrete fracture leaching model more complicated but is a longstanding inspiration in the field of rock engineering [57,58,59]. In the near future, we will improve the current model by taking into account the normal pressure interaction to examine its effects on concrete fracture leaching.

## Figures and Tables

**Figure 1 materials-15-04584-f001:**
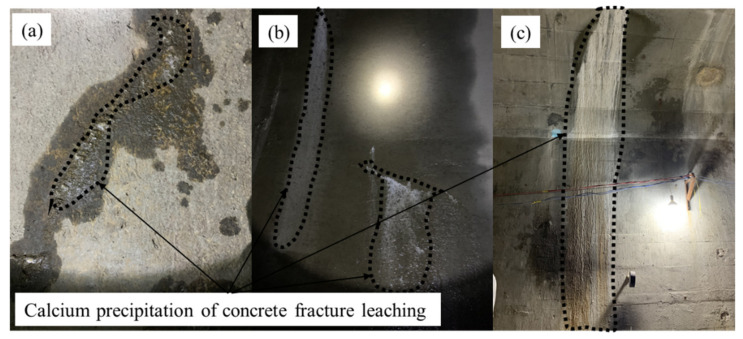
Different leaching degrees of hydration products from fractured regions in concrete (tunnel lining). (**a**) mild, (**b**) moderate and (**c**) severe.

**Figure 2 materials-15-04584-f002:**
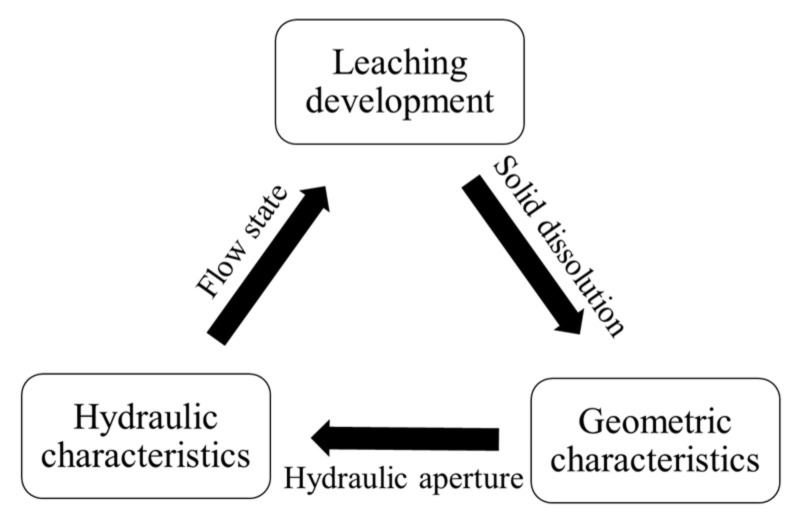
Coupling of the leaching, fracture geometry and hydraulic processes.

**Figure 3 materials-15-04584-f003:**
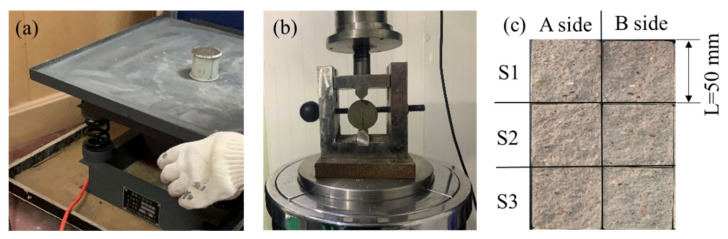
Process of samples preparation. (**a**) Pouring cement samples, (**b**) Brazilian splitting test and (**c**) numbering of fracture surfaces.

**Figure 4 materials-15-04584-f004:**
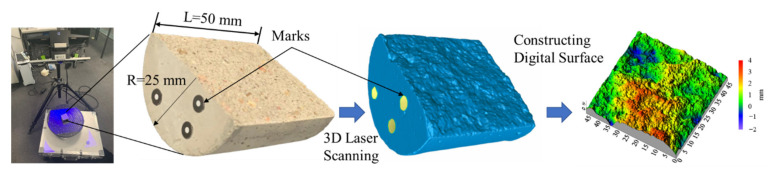
Obtainment process of fracture surfaces by 3D laser scanning.

**Figure 5 materials-15-04584-f005:**
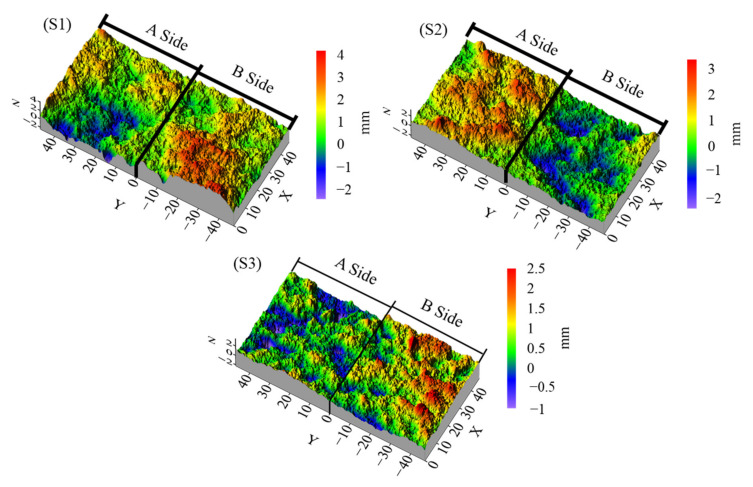
Geometry of concrete samples used in this study.

**Figure 6 materials-15-04584-f006:**
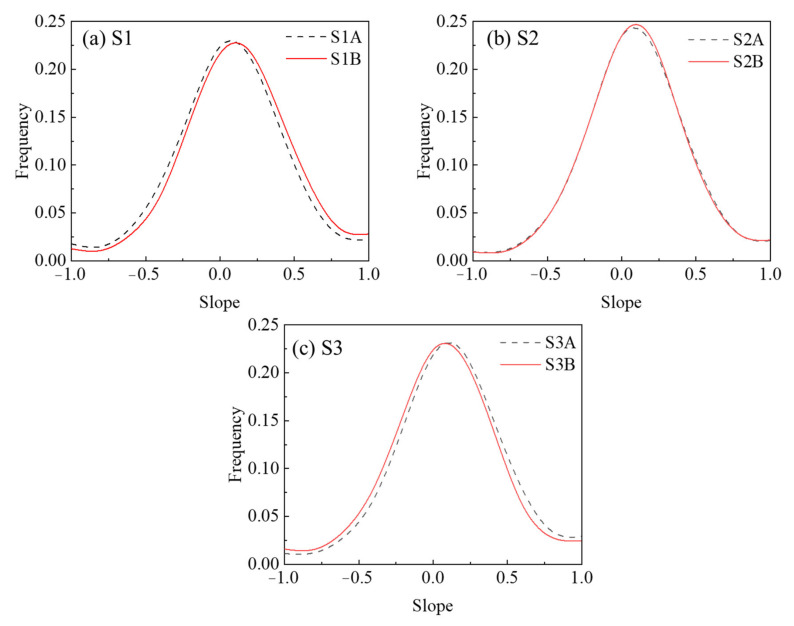
Distribution of surface slope.

**Figure 7 materials-15-04584-f007:**
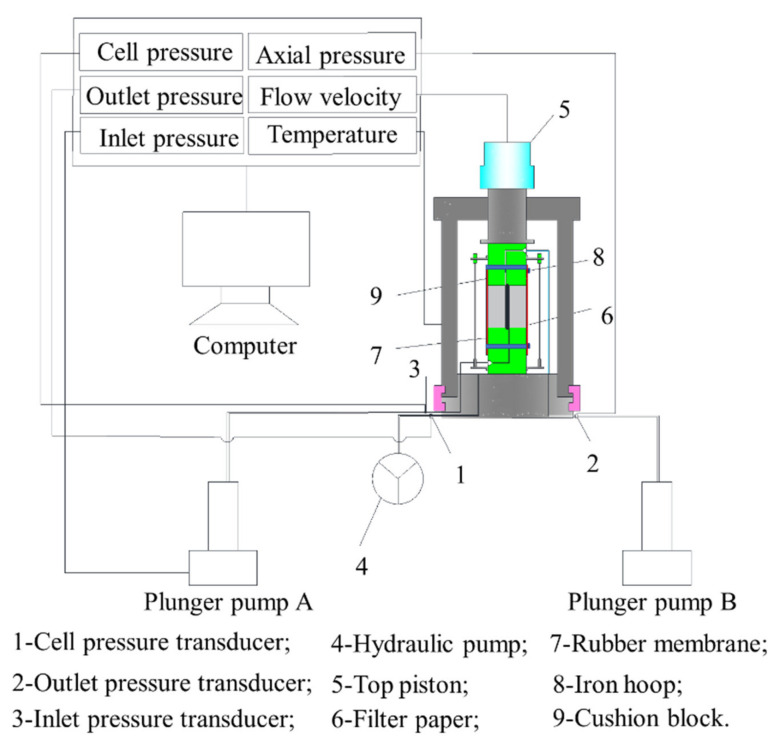
Schematic of experimental setup for permeability experiment.

**Figure 8 materials-15-04584-f008:**
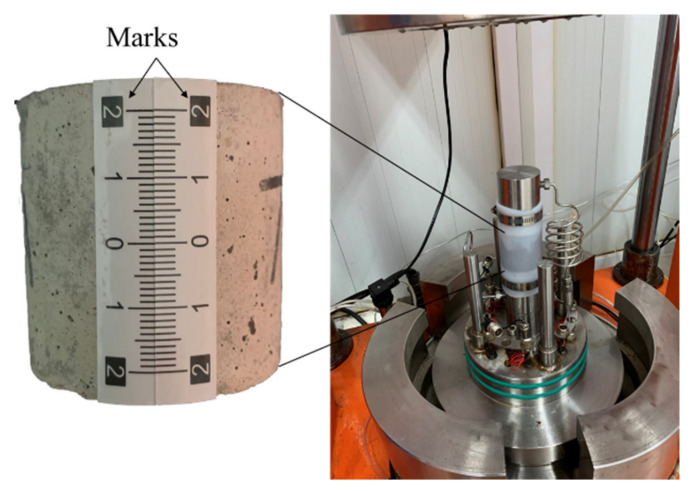
Sample loading.

**Figure 9 materials-15-04584-f009:**
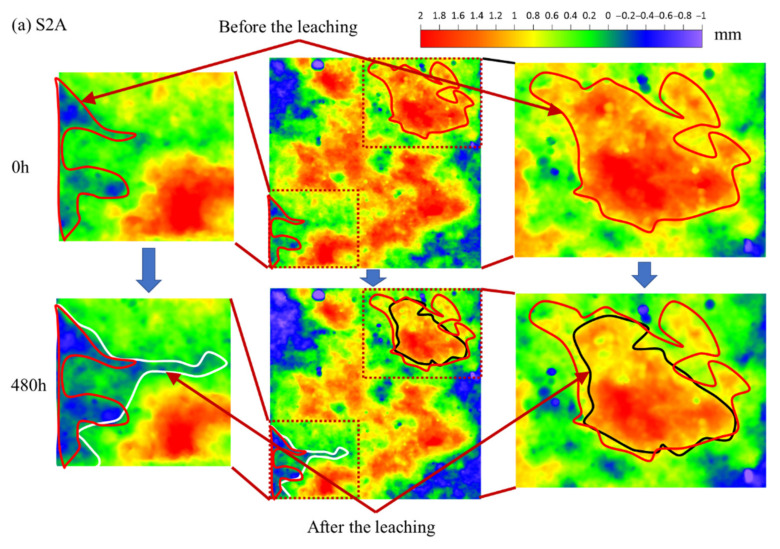
Temporal evolution in fracture surface geometry variation.

**Figure 10 materials-15-04584-f010:**
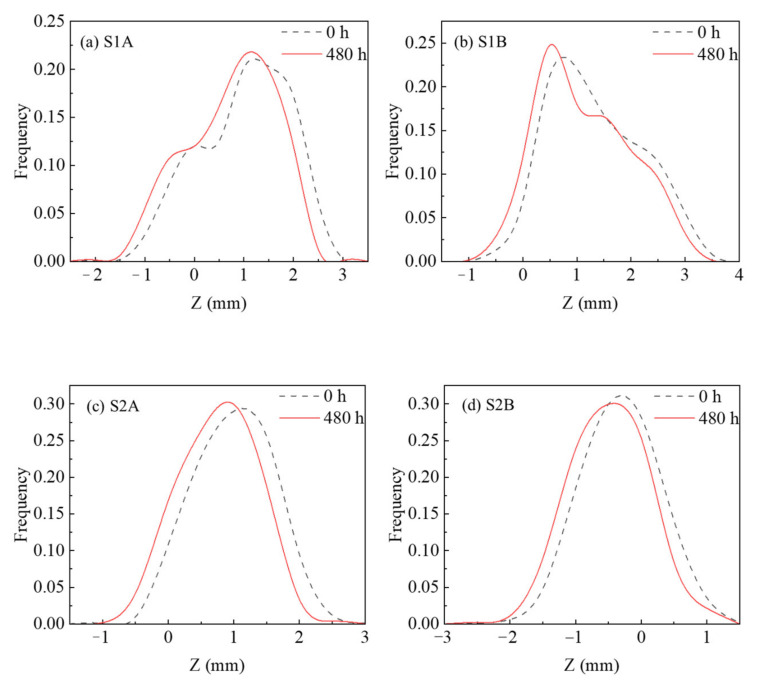
Temporal evolutions in elevation distribution variation.

**Figure 11 materials-15-04584-f011:**
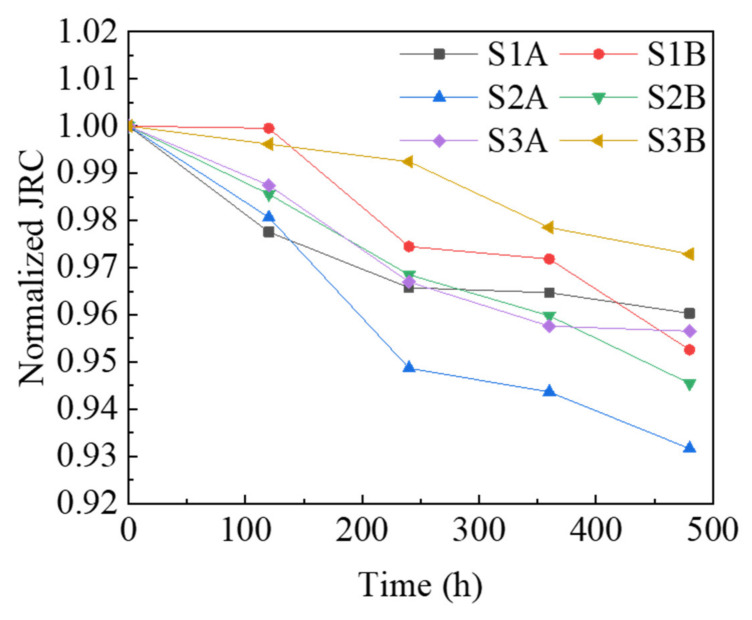
Temporal evolutions in fracture surface normalized JRC variation.

**Figure 12 materials-15-04584-f012:**
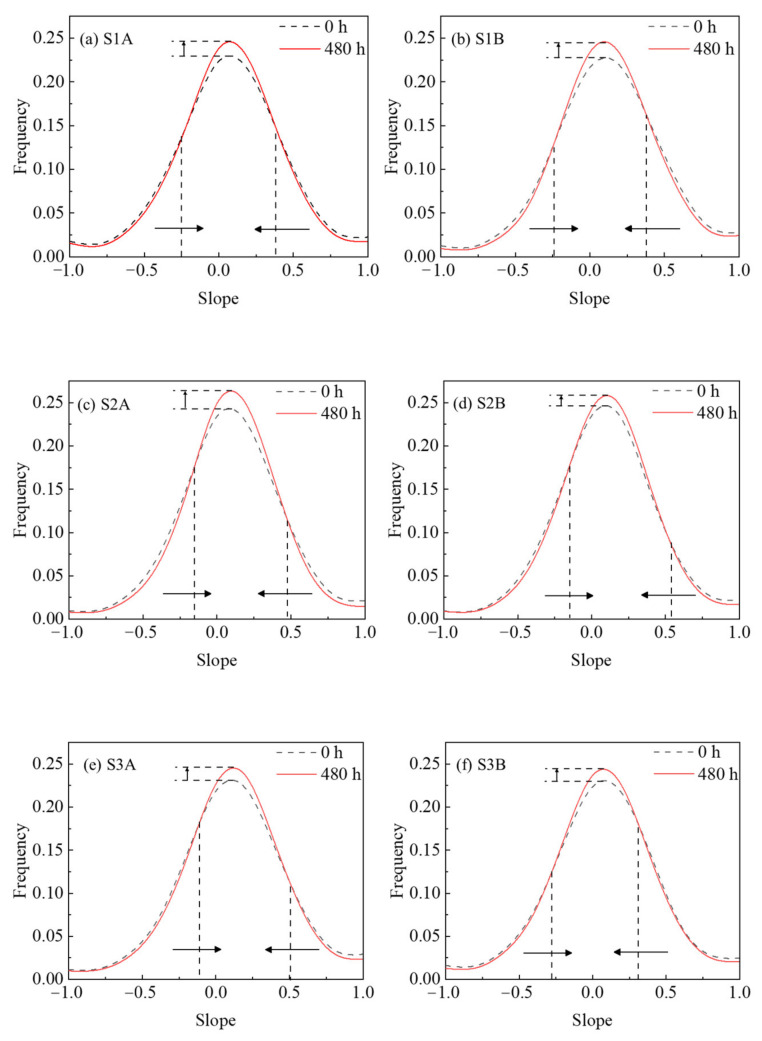
Temporal evolutions in slope distribution variation.

**Figure 13 materials-15-04584-f013:**
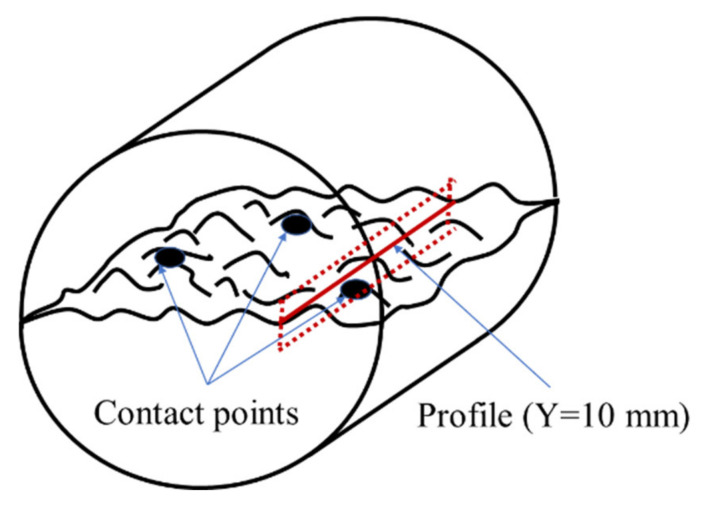
Illustration of fitting the two fracture surfaces and intercepting the profile.

**Figure 14 materials-15-04584-f014:**
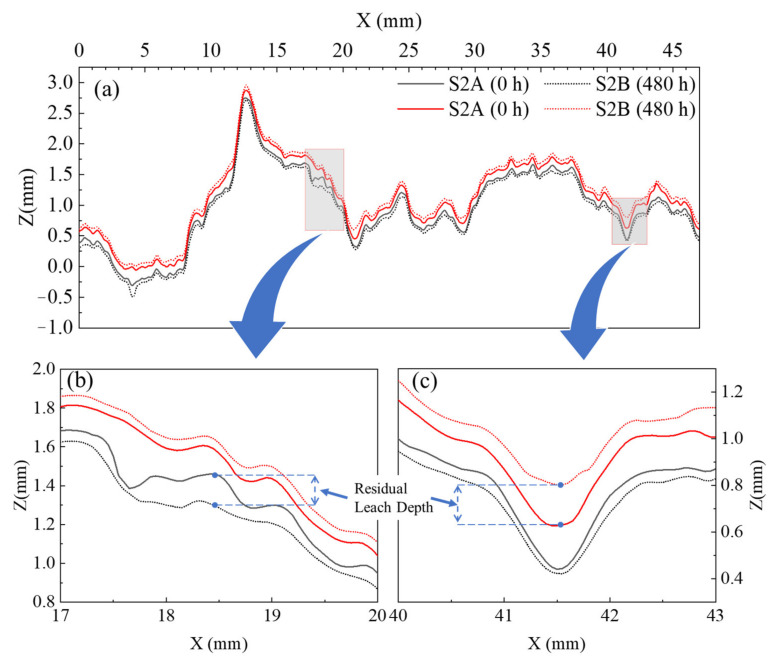
Temporal evolutions in profile variation. (**a**) profile of Y = 10 mm, (**b**) enlarged view of X = 17−20 mm and (**c**) enlarged view of X = 40−43 mm.

**Figure 15 materials-15-04584-f015:**
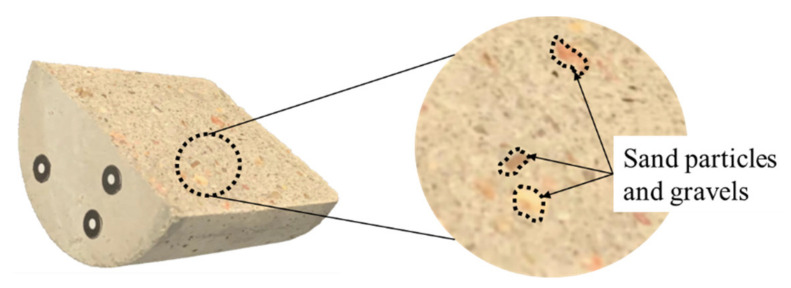
Illustration of sand particles and gravel embedded into the fracture surface.

**Figure 16 materials-15-04584-f016:**
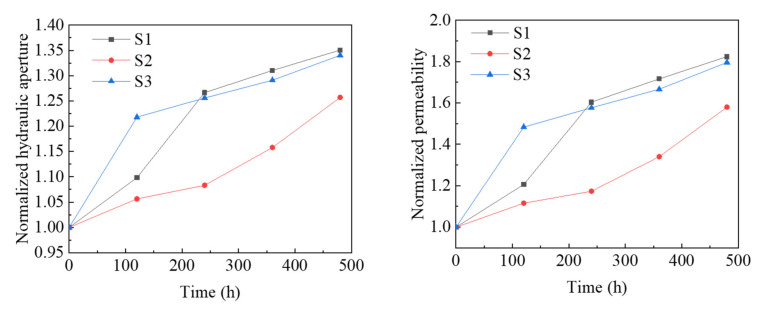
Temporal evolutions in fracture normalized hydraulic aperture (**left**) and permeability (**right**) variation.

**Figure 17 materials-15-04584-f017:**
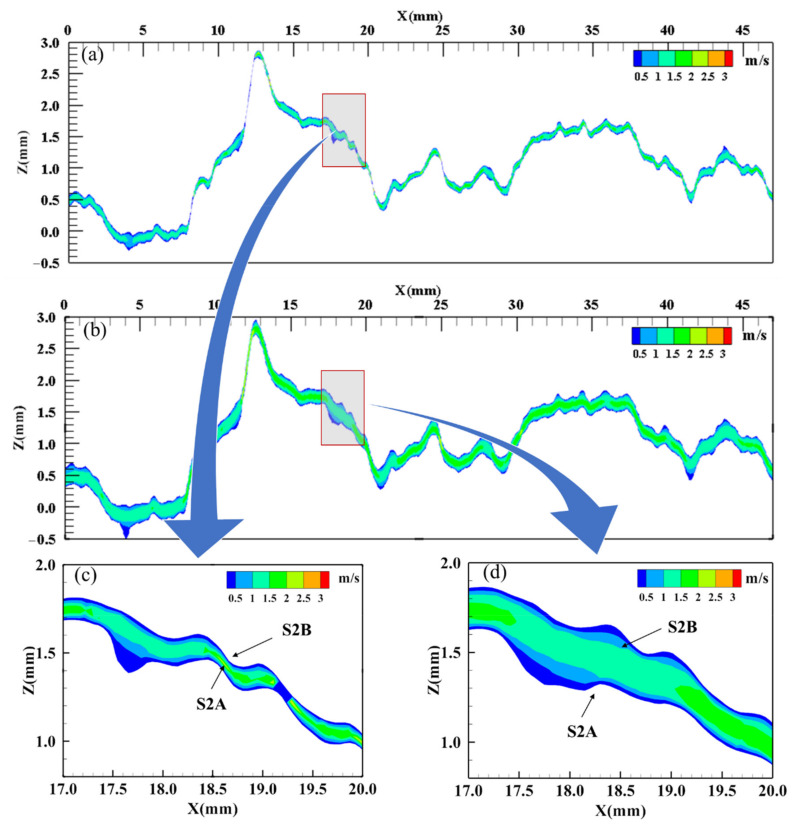
Numerical simulation of fluid flow in actual fracture model by Fluent 16.0. (**a**) Fracture before leaching, (**b**) fracture after leaching, (**c**) enlarged view of X = 17−20 mm before leaching and (**d**) enlarged view of X = 17−20 mm after leaching.

**Figure 18 materials-15-04584-f018:**
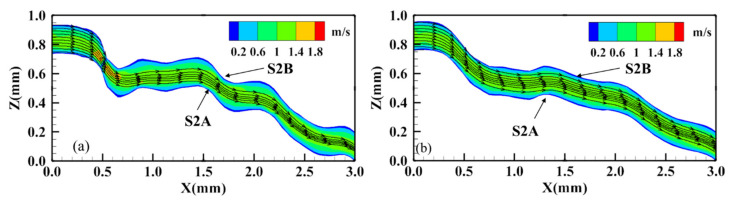
Numerical simulation of fluid flow in idealized fracture model (equal mechanical aperture) by Fluent 16.0. (**a**) fracture before leaching and (**b**) fracture after leaching.

**Figure 19 materials-15-04584-f019:**
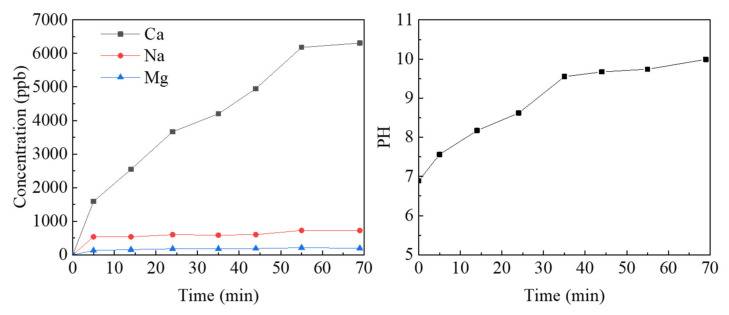
The analysis of chemical composition (**left**) and PH value (**right**) in leaching solution.

**Figure 20 materials-15-04584-f020:**
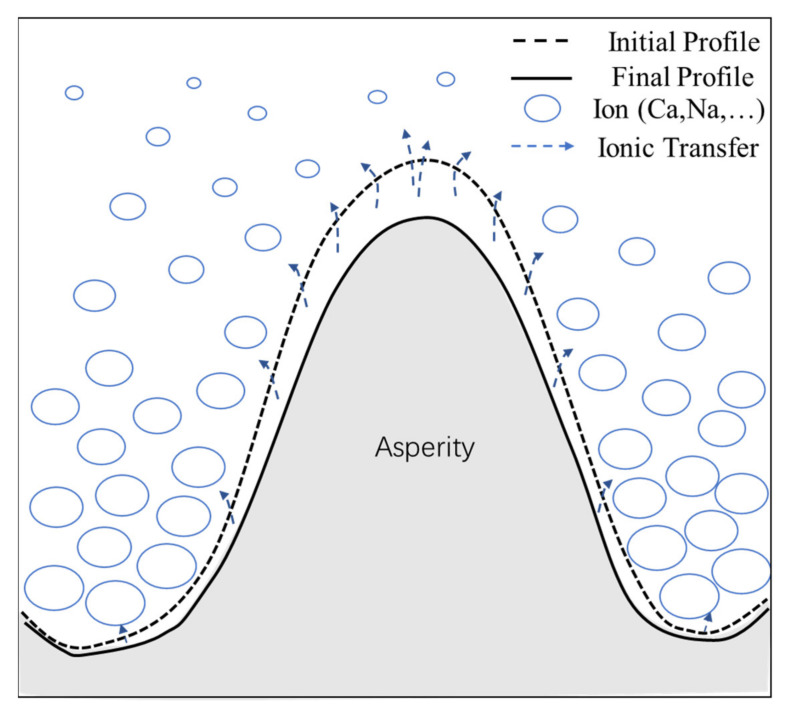
Illustration of concrete rough fracture leaching model.

**Table 1 materials-15-04584-t001:** Initial JRC values for sample surfaces.

Fracture Surface Number	JRC (0 h)
S1A	22.0
S1B	20.5
S2A	20.4
S2B	20.0
S3A	20.4
S3B	21.0

**Table 2 materials-15-04584-t002:** Fracture surface JRC values at different leaching time.

Fracture Surface Number	JRC				
	0 h	120 h	240 h	360 h	480 h
S1A	22.0	21.5	21.3	21.3	21.2
S1B	20.5	20.5	20.0	19.9	19.5
S2A	20.4	20.0	19.3	19.2	19.0
S2B	20.0	19.7	19.4	19.2	18.9
S3A	20.4	20.2	19.8	19.6	19.6
S3B	21.0	20.9	20.8	20.5	20.4

**Table 3 materials-15-04584-t003:** Experimental results of flow rate.

Sample Number	Flow Rate (mL/min)				
	0 h	120 h	240 h	360 h	480 h
S1	3.74	4.96	7.60	8.41	9.22
S2	3.28	3.86	4.17	5.08	6.51
S3	3.00	5.42	5.94	6.46	7.22

**Table 4 materials-15-04584-t004:** Calculated hydraulic aperture at different leaching times.

Sample Number	*b**_n_* (m)				
	0 h	120 h	240 h	360 h	480 h
S1	1.55 × 10^−5^	1.7 × 10^−5^	1.97 × 10^−5^	2.03 × 10^−5^	2.1 × 10^−5^
S2	1.49 × 10^−5^	1.57 × 10^−5^	1.61 × 10^−5^	1.72 × 10^−5^	1.87 × 10^−5^
S3	1.44 × 10^−5^	1.76 × 10^−5^	1.81 × 10^−5^	1.86 × 10^−5^	1.93 × 10^−5^

**Table 5 materials-15-04584-t005:** Calculated permeability at different leaching times.

Sample Number	*k* (m^2^)				
	0 h	120 h	240 h	360 h	480 h
S1	2.01 × 10^−11^	2.42 × 10^−11^	3.22 × 10^−11^	3.45 × 10^−11^	3.66 × 10^−11^
S2	1.84 × 10^−11^	2.05 × 10^−11^	2.16 × 10^−11^	2.46 × 10^−11^	2.9 × 10^−11^
S3	1.73 × 10^−11^	2.57 × 10^−11^	2.73 × 10^−11^	2.89 × 10^−11^	3.11 × 10^−11^

## Data Availability

The data used to support the findings of this study are included within the article.

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
