# Peer review of "Effect of Leaching Behavior on the Geometric and Hydraulic Characteristics of Concrete Fracture"

_materials, 2022, doi:10.3390/ma15134584_

Round 1

Reviewer 1 Report

In this paper, Tthe effect of leaching behavior on the geometric and hydraulic characteristics of concrete fracture has been studied. Few comments are given below. 

-In lines 122-132, and other parts, where the sentences are written not in a solid part, the authors should refine the content and write it as a one paragraph. 

-Overall the manuscript, the equations and sympols should be Word-Type (Microsoft Office), not MathType (i.e., 240-243), so as to appeared for the readers clear and in line with the text. 

-Fig. 19, the text of the axis can be little enlarged like Fig. 15. 

-Fig. 18. the text of the axis in some figures are large while in other figures it is small. They should all be with the same characteristics. 

-In Fig. 13, the legend should be clear. Different line thickness would be good to make it more clear. 

Author Response

Response to Reviewer 1 Comments

Point 1: In lines 122-132, and other parts, where the sentences are written not in a solid part, the authors should refine the content and write it as a one paragraph.

Response 1: Thank you for your valuable comments. Lines 122-132 have been described in segments.

Point 2: Overall the manuscript, the equations and sympols should be Word-Type (Microsoft Office), not MathType (i.e., 240-243), so as to appeared for the readers clear and in line with the text.

Response 2: Thank you for your valuable comments. The equations and sympols have been modified to Word-Type.

Point 3: Fig. 19, the text of the axis can be little enlarged like Fig. 15.

Response 3: Thank you for your valuable comments. Have been modified.

Point 4: Fig. 18. the text of the axis in some figures are large while in other figures it is small. They should all be with the same characteristics.

Response 4: Thank you for your valuable comments. Have been modified.

Point 5: In Fig. 13, the legend should be clear. Different line thickness would be good to make it more clear.

Response 5: Thank you for your valuable comments. Have been modified.

Reviewer 2 Report

The paper has been devoted to study the effect of leaching on the geometric and hydraulic properties of concrete fracture mechanics. In this research, several experimental test have been conducted. 3D laser scanning technique has been employed to evaluate the geometric properties of the concrete after leaching test. The hydraulic characteristics of leached concrete have been also studied using permeability test. The paper has been well-written. The research topic is interesting. The text of the paper is readable however, there are several typo mistakes within the text which should be corrected by the authors. The results obtained by this study is new and original. The authors should discuss the novelty and the main contribution of this research clearly. In addition, the main advantages of the obtained results on the real work should be clarified. In my pinion, after considering the aforementioned minor points by the authors in the revised manuscript, the paper can be recommended for publication. 

Author Response

Response to Reviewer 2 Comments

Point 1: The text of the paper is readable however, there are several typo mistakes within the text which should be corrected by the authors.

Response 1: Thank you for your valuable comments. The typo mistakes within the text have been corrected.

Point 2: The authors should discuss the novelty and the main contribution of this research clearly. In addition, the main advantages of the obtained results on the real work should be clarified.

Response 2: Thank you for your valuable comments. Relevant contents have been briefly explained in the conclusion

Reviewer 3 Report

The paper titled “Effect of leaching behavior on the geometric and hydraulic characteristics of concrete fracture” (Manuscript ID materials-1755696), presents an interesting research study. The manuscript is well written, with Figures being correctly displayed and very informative. Methods and Results provide the necessary support for the discussion section. Conclusions are balanced, well written, and successfully display the major outlines of this paper. Just a few suggestive observations to further improve your manuscript:

In figure 5: I know that the size of the cylinder is mentioned in the methodology, but it would be better for the reader to have this info in a form of scale bar or just a brief comment in the figure legend. Please apply to all figures with cylinder.

In figure 16: if these particles are bigger than 2mm you should characterize them as gravels.

Therefore, I suggest minor revision prior to publication in the Journal of “Materials”.

Author Response

Response to Reviewer 3 Comments

Point 1: In figure 5: I know that the size of the cylinder is mentioned in the methodology, but it would be better for the reader to have this info in a form of scale bar or just a brief comment in the figure legend. Please apply to all figures with cylinder.

Response 1: Thank you for your valuable comments. The size of the cylinder has been marked in relevant figures.

Point 2: In figure 16: if these particles are bigger than 2mm you should characterize them as gravels.

Response 2: Thank you for your valuable comments. Have been modified.

Reviewer 4 Report

This is an interesting and potentially important study. Several comments are given for improving the paper.

1. Some of the key terms need refinement. At the start of the abstract, "the leaching of fractures" can be "the leaching of material from concrete fracture surfaces" or something like that.

"leached-geometric-hydraulic process" would not be understood at the start of the abstract.

"investigate the mechanisms" seems better than "investigate the law and mechanism". The research does not come close to establishing a "law".

"are increased monotonically" with respect to what? Degree of leaching?

Still in the abstract "development of leaching on the concrete fractures" can be "development of leaching local to concrete fracture surfaces" or something like that.

Likewise in the paper (for example on line 48) "leaching of fractures in concrete" can be "leaching of hydration products from fractured regions in concrete". Similar remark on the title of Figure 1.

Line 62 "coupled leached-geometric-hydraulic process" might be understood by readers but it is awkward expression. Some rethinking is needed.

Figure 2 needs some modifications. Use "leaching development". The title can be "Coupling of the leaching, fracture geometry and hydraulic processes."

Figure 4 is not needed. It can be explained by text.

3D laser scanning is an established technology. The following text is not needed and can be removed. “This technology inverses the spatial coordinates (x, y, z) of each reflective point on the fracture surface by emitting and receiving laser beam, and reconstructs the fracture surface through 3D processing software (such as Geomagic Studio). Therefore, the geometry of the fracture surface and its changes can be obtained accurately with sufficient scan interval and accuracy.”

In Figure 11 it is difficult to distinguish between the two line styles or colors. More contrast is needed between the line styles. Same comment applies to Figure 13.

The flow simulations in Figure 18 and 19 are possibly misleading since roughness in the out of plane direction will affect flow behavior.

In the conclusions how does this research help “avoid.. the development of concrete fractures”?

Near the end of the introduction, “improve” is preferred over “perfect”.

Assistance is needed to improve writing and presentation quality.

Author Response

Response to Reviewer 4 Comments

Point 1: Some of the key terms need refinement. At the start of the abstract, "the leaching of fractures" can be "the leaching of material from concrete fracture surfaces" or something like that.

Response 1: Thank you for your valuable comments.  "the leaching of fractures" has been modified to "the leaching of material from concrete fracture surfaces".

Point 2: "leached-geometric-hydraulic process" would not be understood at the start of the abstract.

Response 2: Thank you for your valuable comments. "leached-geometric-hydraulic process" have been modified to "coupling of the leaching, fracture geometry and hydraulic processes."

Point 3: "investigate the mechanisms" seems better than "investigate the law and mechanism". The research does not come close to establishing a "law".

Response 3: Thank you for your valuable comments. "investigate the law and mechanism" has been modified to "investigate the mechanisms".

Point 4: "are increased monotonically" with respect to what? Degree of leaching?

Response 4: Thank you for your valuable comments. "are increased monotonically" has been modified to "are increased monotonically with leaching time ". the whole sentence in abstract has been modified to "Then the hydraulic characteristics are investigated by permeability tests, and it is found that the fracture hydraulic aperture and permeability are increased monotonically with leaching time. "

Point 5: Still in the abstract "development of leaching on the concrete fractures" can be "development of leaching local to concrete fracture surfaces" or something like that.

Response 5: Thank you for your valuable comments. "development of leaching on the concrete fractures" has been modified to "development of leaching local to concrete fracture surfaces".

Point 6: Likewise in the paper (for example on line 48) "leaching of fractures in concrete" can be "leaching of hydration products from fractured regions in concrete". Similar remark on the title of Figure 1.

Response 6: Thank you for your valuable comments. "leaching of fractures in concrete" has been modified to "leaching of hydration products from fractured regions in concrete".

Point 7: Line 62 "coupled leached-geometric-hydraulic process" might be understood by readers but it is awkward expression. Some rethinking is needed.

Response 7: Thank you for your valuable comments. "coupled leached-geometric-hydraulic process" has been modified to "coupling of the leaching, fracture geometry and hydraulic processes"

Point 8: Figure 2 needs some modifications. Use "leaching development". The title can be "Coupling of the leaching, fracture geometry and hydraulic processes."

Response 8: Thank you for your valuable comments. "leached development" has been modified to "leaching development".

Point 9: Figure 4 is not needed. It can be explained by text.

Response 9: Thank you for your valuable comments. Figure 4 has been deleted.

Point 10: 3D laser scanning is an established technology. The following text is not needed and can be removed. “This technology inverses the spatial coordinates (x, y, z) of each reflective point on the fracture surface by emitting and receiving laser beam, and reconstructs the fracture surface through 3D processing software (such as Geomagic Studio). Therefore, the geometry of the fracture surface and its changes can be obtained accurately with sufficient scan interval and accuracy.”

Response 10: Thank you for your valuable comments.”This technology inverses the spatial coordinates (x, y, z) of each reflective point on the fracture surface by emitting and receiving laser beam, and reconstructs the fracture surface through 3D processing software (such as Geomagic Studio). Therefore, the geometry of the fracture surface and its changes can be obtained accurately with sufficient scan interval and accuracy.” has been deleted.

Point 11: In Figure 11 it is difficult to distinguish between the two line styles or colors. More contrast is needed between the line styles. Same comment applies to Figure 13.

Response 11: Thank you for your valuable comments. The two lines in the figure have been described with black dashed lines and red solid lines.

Point 12: The flow simulations in Figure 18 and 19 are possibly misleading since roughness in the out of plane direction will affect flow behavior.

Response 12: Thank you for your valuable comments. We share your view that " roughness in the out of plane direction will affect flow behavior. " But this section is to study how changes in roughness affect flow characteristics. Therefore, the fracture contour is randomly selected to simulate the flow. The roughness data in the previous Section 3.1 of this paper are all along the X direction, so the two profiles selected in the flow simulation are also in the X direction. Although the roughness in the out of plane direction will affect the flow behavior, it is not analyzed in this section.

Point 13: In the conclusions how does this research help “avoid.. the development of concrete fractures”?

Response 13: Thank you for your valuable comments. “avoid.. the development of concrete fractures” will be studied in future, and this article does not address how to do it. Therefore, the sentence “This is significant in the underground engineering field, especially for avoiding and predicting the development of concrete fractures under rich groundwater environment” has been deleted.

Point 14: Near the end of the introduction, “improve” is preferred over “perfect”.

Response 14: Thank you for your valuable comments. “perfect” has been modified to “improve”.

Point 15: Assistance is needed to improve writing and presentation quality.

Response 15: Thank you for your valuable comments. Sorry for bringing you this unpleasant reading experience. In the new manuscript, we tried our best to improve the language and expressions: We had the English in the manuscript thoroughly checked and edited for language and form. An expert whose first language is English was invited to revise and polish the whole article.

Reviewer 5 Report

Dear Authors,

thank you very much for your important and well prepared article.

The topic of leaching behaviour in combination with concrete fracture is very interesting and definitely important.

Here a few observations for improvement, but I conclude that the article is almost suitable for publication:

The introduction is rather short, but okay.

I recommend improving the quality by adding information on degradation in aggressive environments, e.g:

10.24425/amm.2019.131099

The description of samples, material and process is fine.

The results are presented clearly.

Conclusions are adequate but long - need to be more precise and concise. 

Regards

Author Response

Response to Reviewer 5 Comments

Point 1: I recommend improving the quality by adding information on degradation in aggressive environments, e.g: 10.24425/amm.2019.131099

Response 1: Thank you for your valuable comments. The information on degradation in aggressive environments has been added. The recommended literature has been cited in the text.

Point 2: Conclusions are adequate but long - need to be more precise and concise.

Response 2: Thank you for your valuable comments. The conclusions have been modified.
